# Management of the Secondary Spontaneous Pneumothorax: Current Guidance, Controversies, and Recent Advances

**DOI:** 10.3390/jcm11051173

**Published:** 2022-02-22

**Authors:** George William Nava, Steven Philip Walker

**Affiliations:** Academic Respiratory Unit, Southmead Hospital, Bristol BS10 5NB, UK; steven.walker@nbt.nhs.uk

**Keywords:** secondary spontaneous pneumothorax, persistent air leak, autologous blood patch

## Abstract

Secondary spontaneous pneumothorax (SSP) is a medical emergency where the lung collapses in the presence of underlying chronic lung disease. It is the commonest cause of spontaneous pneumothorax and results in significant breathlessness, higher morbidity, mortality, and longer hospital admissions than with patients with pneumothoraces and no underlying lung disease. This article explores the current guidance, controversies, and recent advances in the management of this condition.

## 1. Introduction

A spontaneous pneumothorax (SP) describes the movement of air from the lung into the pleural cavity in the absence of trauma. A secondary spontaneous pneumothorax (SSP), as opposed to a primary spontaneous pneumothorax (PSP), refers to a SP in a patient with underlying chronic lung disease such as chronic obstructive pulmonary disease (COPD), interstitial lung disease (ILD), cystic fibrosis or lung cancer, or acute conditions, such as Pneumocystic jirovecii pneumonia (PJP) or COVID-19 pneumonitis [1,2]. SSPs are more common than PSPs, with recent English SP epidemiological data demonstrating over 60% are due to SSPs. The outcomes for SSPs are worse than PSPs, with higher recurrence rates and longer hospital lengths of stay [3,4]. Despite being commoner, the evidence based for SSPs is smaller than that for PSPs. Fortunately, new randomised studies focused on the initial management and recurrence prevention in SSP have been published. In this article, we will review existing guidelines, describe recent advances in management, and discuss controversies in the field.

## 2. Overview of Management of Secondary Spontaneous Pneumothorax

### 2.1. Current Management Guidelines

In general, SSPs occur in older patients, often with more accompanying health conditions than patients with PSPs [5]. Patients are more frequently hypoxic [6], more likely to develop persistent air leaks [6,7] and have a higher risk of recurrence and inpatient mortality than PSPs [3,5,8]. As a result, management guidelines have separated the management of PSPs and SSPs, with a preference towards intervention on SSP pathways.

The British Thoracic Society (BTS) guidance [9] defines SSP as a SP occurring in a patient with underlying lung disease (Figure 1) or in a patient above the age of 50 with a significant smoking history, citing reduced success rates with aspiration and the high probability of underlying disease. The American College of Chest Physicians (ACCP) 2001 Delphi Consensus Statement [10] on the management of SP do not include an age criteria in their definition. The BTS guidance outlines the initial management of an SSP, in the absence of haemodynamic compromise, with three options:Conservative inpatient management if the SSP is <1 cm at the hilum.14–16 gauge needle aspiration (NA) and admission for observation if the SSP is 1–2 cm at the hilum.<14-French chest tube drainage (CTD) if the patient is unstable, breathless, if the pneumothorax is >2 cm at the hilum, or >1 cm at the hilum after an attempted aspiration.

If haemodynamic compromise is present, then urgent needle decompression with a 14-gauge cannula in the 2nd intercostal space mid-clavicular line or 4th/5th intercostal space mid-axillary line followed by CTD insertion is recommended.

Some early studies suggest that oxygen supplementation may help accelerate the resolution of a pneumothorax [11,12], however care must be taken in the context of a SSP as many patients will be at risk of type 2 respiratory failure.

### 2.2. Initial Management: Needle Aspiration [NA] vs. Chest Tube Drainage [CTD]

NA involves inserting a small-bore cannula into the chest and aspirating air with a syringe before removing the cannula once all the air is removed. It is a simple procedure to rapidly relieve symptoms and has been associated with lower length of hospital stays with fewer complications in a randomised trial in patients with PSP [13]. The shortcoming is that it fails if there is a persistent visceral air leak. Early data showed that NA of an SSP is less successful than for PSP [14,15], hence it does not play a significant role in the BTS guidance [9] and is not recommended for SSP under any circumstance by the ACCP [10].

Recent data suggest that there might be a more prominent role for NA in SSPs. In a multicentre study in Norway of patients with SP [16], a subgroup of 48 patients with SSP were randomised to CTD or NA, with a repeat NA performed if the first did not relieve breathlessness and result in a pneumothorax that was <20%, as defined by Rhea calculation. Those who had NAs had an immediate success rate of 59% and a reduced length of hospital stay compared with CTD (2.5 vs. 5.5 days). There were no complications from the NA procedures, but 15 complications from CTDs, including a death related to an empyema. Of note, the success rate of the second NA was nearly 50% despite the failure of the first. Whilst these results are promising and support a significant deviation from current BTS guidance, the SSP population in this study was not well defined and the number of participants was small.

If CTD is required, ACCP guidance suggests large bore chest tubes in most circumstances and small-bore tubes (<14 F) are used only if the pneumothorax is small). BTS guidance challenges this and suggests small bore tubes are appropriate as the first-line option due to retrospective evidence showing similar performance of 10–14 F to 20–28 F tubes in SSP [17]. This is supported by a randomised controlled trial (RCT) of 22 patients that showed equivalence of 14 F and 30 F chest tubes in patients with SSP [18].

### 2.3. Conservative Management

There has been a lot of interest in a noninvasive, conservative approach to SP, after a recent RCT of conservative versus interventional management for moderate-large PSPs demonstrated non-inferiority for a primary outcome of radiological resolution at eight weeks. Conservative care also demonstrated improvements in 12-month recurrence rates as well as reduced adverse outcomes, with a modest (15%) reintervention rate [19]. However, there have been no prospective studies that have trialed conservative management of SSPs. A recent case report describes successful management of two SSPs despite initial breathlessness [20], and a retrospective cohort study describes the successful conservative management of 25 SSPs that were greater than 1 cm at the hilum [21]. These are non-randomised observational data, with smaller SSPs in the conservatively managed group and no explanation of the clinician’s decision-making process. Nevertheless, they describe a cohort of patients that were successfully managed against current guidelines, though prospective trials must be undertaken before this can be recommended.

### 2.4. Ambulatory Management

Ambulatory management is achieved with the insertion of a chest tube with a one-way flutter valve at the external end (Figure 2). A patient can mobilise freely without needing to manage large underwater seal chambers. Devices can be attached to the end of a conventional chest tube, or valves can be incorporated into purpose-designed devices. RCTs of ambulatory devices in PSPs have shown reduced lengths of stay [22,23,24], though with a higher rate of serious adverse events in the largest study [24].

Retrospective data describe a high (78%) success rate of outpatient management with ambulatory drainage of selected SSPs [25]. In a multicentre prospective RCT in the United Kingdom [26], 41 patients with SSPs were randomised to ambulatory management or standard CTD. This study found an initial shorter length of stay in the ambulatory group but no difference in the total length of hospitalisation due to a high readmission rate with the ambulatory devices. The study used both flutter valves attached to 12 F chest tubes and an integrated device with an 8 F catheter, with a significantly higher failure rate of the integrated devices. It is considered that the 8 F catheters are too small to manage the high rate of air leaks that are more common in SSP, or to cope with the higher secretion burden, hence the higher rates of surgical emphysema in this group. Ambulatory management remains an appealing option to avoid unnecessary inpatient stays. Current data suggest a 12 F chest tube with a flutter valve might be most successful, though further evidence is required to guide the candidate selection.

### 2.5. Management Options for Persistent Air Leaks (PALs)

A PAL describes the ongoing leak of air from the lungs into the pleural space after chest drain insertion [9]. BTS guidance suggests a referral to the thoracic surgical team at 48 h, however, many patients with SSP are not considered appropriate for a surgical procedure, and alternative strategies have been explored. These include:Persistence with CTDThe application of suctionChemical pleurodesisAutologous blood patch (ABP)Endobronchial/intrabronchial valves.

#### 2.5.1. Persistence with CTD

PALs are more common and less likely to resolve with SSPs than PSPs. However, observational data suggest 61% of patients with SSPs and PALs resolve at seven days, and 81% will resolve by 14 days without further intervention [27]. Keeping a patient in hospital with a chest tube for weeks, without the guarantee that the leak will resolve, is a psychological and physical burden, increases the risk of health complications, and poses a financial burden to the health service. Exploring options to hasten the resolution is therefore important.

#### 2.5.2. Suction

Suction refers to the application of negative pressure to the drainage system to accelerate the removal of air from the pleural space and promote air leak closure through apposition of the visceral and parietal pleurae. Studies have not shown a benefit of suction for pneumothorax resolution [28]. Interventional management of PSPs is associated with an increased recurrence rate compared with conservative management. This is thought to be due to the active expansion of the lung impairing healing of the lung defect—an effect that suction would potentially exaggerate. Newer digital air leak systems (Figure 3) allow continuous real-time air leak data, which as well as providing suction, may help predict the trajectory of a PAL [29,30].

#### 2.5.3. Chemical Pleurodesis

Chemical pleurodesis refers to the instillation of an agent down a chest tube to promote inflammation and a fusion of the visceral and parietal pleurae. This requires the lung to be fully inflated for the pleurae to be in apposition. This promotes the sealing of the air leak but will also contribute to recurrence prevention. Different agents have been trialed for medical pleurodesis, however, there are few prospective controlled trials of these. Two small studies (one controlled) suggest a poor effect of intrapleural tetracycline on PALs [31,32]. A more recent study demonstrated successful pleurodesis in 5/6 patients with talc slurry administration for PAL with an SSP [33]. However, the median time for the air leak to stop post-pleurodesis was 12 days, and in the absence of a control group, it is difficult to say that the talc had any effect. Alternative agents, such as glucose and OK-432, have been used [34].

#### 2.5.4. Autologous Blood Patch (ABP)

It has been postulated that the instillation of small volumes (50–120 mL) of autologous blood into the pleural space through a chest drain may form a clot over the broncho-pleural fistula, accelerate the healing of a PAL and possibly provoke a pleurodesis reaction. There have been, however, ongoing concerns regarding the risk of the pleural infection with introducing blood into the pleural space. Small non-randomised studies that included post-operative patients have demonstrated good success rates, with a literature review from 2010 of 109 patients with a PAL following SP reporting a 91% success rate with ABPs [35,36,37,38,39,40]. There have been two RCTs of patients with SSPs which compared ABPs with conservative management, which demonstrated a significantly reduced time for PAL resolution [41,42]. Whilst these studies are promising, their numbers are small with only 47 and 44 patients in the respective studies. Ibrahim et al. randomised 47 patients to 50 mL of autologous blood at day 3 post-CTD and compared this to free drainage up until day 10, at which point the patients received APB if there was a PAL [41]. Both arms had high rates of ABP administration, with all 23 in the early ABP cohort receiving ABP and 16/24 (67%) of the usual care receiving ABP. The time to air leak seal was 5.4 (±1.3) days and 10.5 (±3.1) days in the early ABP versus standard care arm. Pleural infection rates were non-significantly different in both cohorts, at 8.7% and 16.7% in early ABP and standard care. Cao et al. randomised 44 patients at day 7 post-CTD to either 0.5 mL/kg, 1 mL/kg, 2 mL/kg autologous blood or 1 mL/kg of normal saline as a control [42]. The study defined success as cessation of air leak at day 20 post-procedure, with a higher proportion of patients meeting these criteria at 1 mL/kg and 2 mL/kg (82% success in both) compared to either 0.5 mL/kg of ABP or saline (27% and 9%, respectively). There were no reported cases of pleural infection during the study. It must be noted that in both these randomised studies, the majority of patients received more than 1 ABP administration, with 58% (27/47) and 76% (25/33) receiving two or administrations [41,42]. In summary, whilst ABP does appear to accelerate the resolution of a PAL, it requires serial administrations, and the overall duration of the PAL remains high in the studied patients.

#### 2.5.5. Endobronchial Valves (EBVs)

EBVs or intrabronchial valves, are placed with a flexible bronchoscopic procedure. They are positioned in lobar, segmental or subsegmental bronchi and allow air to move out of the lung, but not into it. They are principally used for lung volume reduction therapy for emphysema [43], however, they have been utilised in other settings, such as post-lung resection PAL and in SP. Sequential endoscopic balloon occlusion of bronchi allows the localisation of an air leak. Placement of an EBV will stop the air leak and allow time for the defective area of the lung to heal. Due to collateral ventilation, the location of the leak may not be identifiable, and in this case, the procedure would need to be abandoned.

A retrospective case series of 37 patients with an SP and PAL, but who had either refused surgery or were deemed not suitable for surgery, were offered EBVs. The median time from chest tube to bronchoscopy was 25 days. The location of the air leak could not be identified in 17 (46%) patients and of the remaining 20, only 8 (40%) were successful in stopping the air leak and allowing the chest drain to be removed [44]. A retrospective study in the US reported success in 48/60 (80%) patients who received EBVs for PAL in the context of SP or iatrogenic pneumothorax [45]. In this latter study, the median time from chest tube insertion to EBV insertion was 10 days, so spontaneous resolution of the air leak might explain some of the differences in success rates between these studies.

EBV insertion for PAL is attractive since it is less invasive than surgery. In the absence of large prospective randomised controlled trials, patient selection is difficult, and the concern remains that delaying surgical management increases the risk of pleural infection and the need for more extensive surgical procedures [46]. Alternative endobronchial therapies have been trialed, such as in a three-armed RCT in China, which included 150 patients with SSPs and a PAL >7 days [47]. They randomised patients to CTD, localised endobronchial ABP injection with thrombin or an endobronchial silicon spigot bronchial occlusion device. The success rate of the intervention was 60% in the CTD arm and 80–85% in the endobronchial intervention arms, though patients were excluded from the analysis if their leak could not be localised. These findings are yet to be replicated.

### 2.6. Surgical Management for PAL

Surgical management of an SSP involves either local anaesthetic medical thoracoscopy, VATS (video-assisted thoracic surgery) or open thoracotomy, with a resection of a section of lung or pleura, and/or a pleurodesis procedure. The choice of surgical technique is reviewed in an accompanying article [48]. At 48 h, it is advised to discuss a PAL with the thoracic surgeons, though the choice of this period is somewhat arbitrary [9]. This advice cites a prospective clinical audit of 42 patents referred to a single surgical unit that showed a higher rate of thoracotomy, rather than the less invasive VATS, in those referred later in their presentation [46]. It also refers to a study that showed a SSP healing rate between 2–10 days of 25% in a retrospective analysis of 8 patients with SSPs [7]. These are at odds with data mentioned earlier in this review that suggest a 50% resolution rate of PAL between day 7 and 14 [27].

SSP surgical management is associated with a mortality rate of up to 5% [49,50,51], though these were single centre retrospective studies analysing data from >10 years ago with no comparator groups of conservatively managed patients. Post-operative mortality is higher in patients with ILD compared with those with COPD [49,52]. In the absence of control groups, it is difficult to know whether this relates to the inherent poor prognosis of those with ILD and SSP, or the surgical management. There is a need for prospective randomised trials to compare medical and surgical management of SSPs and to clarify the optimal timing of intervention.

### 2.7. Recurrence Prevention

BTS guidance suggests recurrence prevention should be offered to patients after their second SSP [9] whereas ACCP guidance suggests it should be after the first [10]. Recurrence rates with conservative management lie between 25–50% and fall to 10–25% after medical pleurodesis and 0–10% after surgical intervention [34,53]. VATS procedures have a low rate of complications but a higher rate of recurrence than open surgical procedures due to the increased chance of missing blebs and the reduced mechanical pleurodesis effects of the procedure [48]. Though there are no head-to-head trials comparing all combinations of intervention, it is advised that the medical approach should be reserved for those who are not fit enough for or who refuse surgery [9,34].

In a retrospective analysis of admissions across the United States of America, same-admission recurrence prophylaxis (including medical pleurodesis, VATS or open surgery) when compared with no further intervention, was associated with a lower rate of recurrence, readmission and odds of mortality both in-hospital and at follow-up [54]. Causality cannot be assessed in this study, and there is invariably selection bias that will influence who was chosen to undergo prophylaxis procedures. Ultimately, the decision to choose recurrence prevention strategies should involve a conversation of the risks and benefits with the patient involved.

### 2.8. Phenotyping SP: Can We Risk Stratify Better?

We currently characterise patients with SP as either PSP or SSP, based on presence of underlying lung disease or age and smoking history. This distinction is centred on existing guidelines which recommended different management pathways based on the reported incidence of PAL and hence the presumed likelihood of treatment failure with conservative care or NA. However, this distinction is not a sensitive predictor of treatment failure, with a failure rate of an initial NA in PSP of 44% (94/212) in clinical trials [16,55,56,57,58]. Thelle et al. [16] suggested for the first time that NA may be as successful in patients with SSP as PSP. Additionally, the current definition of the SSP group includes patients with a range of lung diseases, with differing reported outcomes and treatment success. Patients with ILD who suffer from a pneumothorax have more prolonged air leaks and a higher mortality that those with emphysema [59]. Observational studies suggest a higher failure rate of surgical management of PALs in patients with ILD than COPD [52].

Accordingly, the risk of PAL appears to vary as much within the two cohorts as between them. A 50-year-old patient who smokes may have more in common with 20-year-old patient with PSP, than an 80-year-old with advanced ILD. The paucity of prospective data about SSPs in patients with different underlying lung diseases, especially assessing responses to treatment modalities, makes it challenging to phenotype patients and recommend evidence-based individual management strategies. No radiological studies have convincingly predicted risk of PAL, whilst CT studies using blebs scores to predict recurrence risk provide conflicting results [60]. More research is needed to improve phenotyping SP patients, either with advanced radiological techniques such as lung density assessments or digital air leak monitoring [30,61].

## 3. Conclusions

The management of SSP can be challenging with little high-quality evidence to guide treatment. Recent publications of RCTs in this patient cohort have gone some way to address this, however much more is needed. Further interventional RCTs and improved risk stratification models are required to inform patient centered care and improve outcomes.

## Figures and Tables

**Figure 1 jcm-11-01173-f001:**
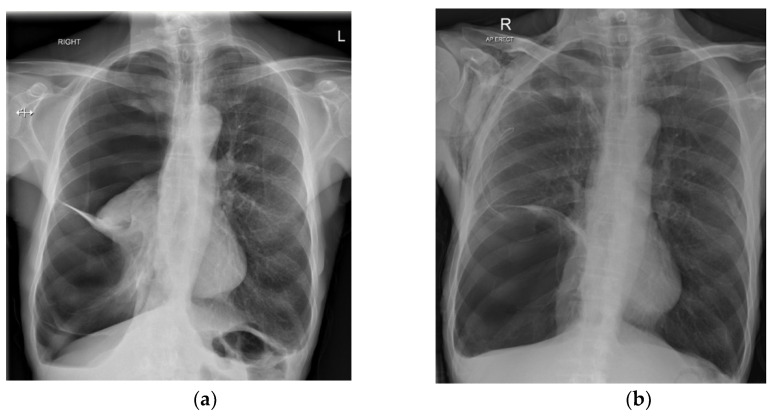
(**a**) A right-sided SSP with evidence of emphysema in the contralateral lung. (**b**) A large bulla evident after drainage and lung reinflation (R, right; L, left).

**Figure 2 jcm-11-01173-f002:**
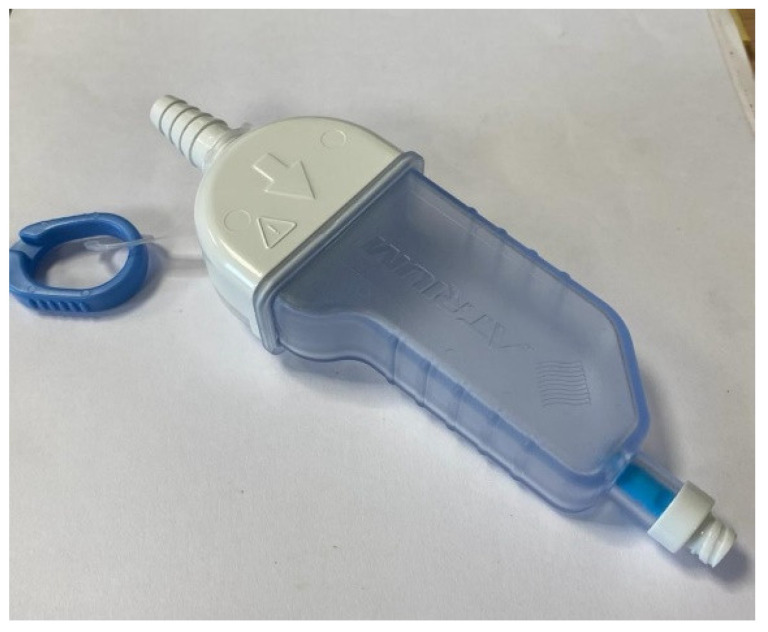
An Atrium Pneumostat—one-way flutter valve.

**Figure 3 jcm-11-01173-f003:**
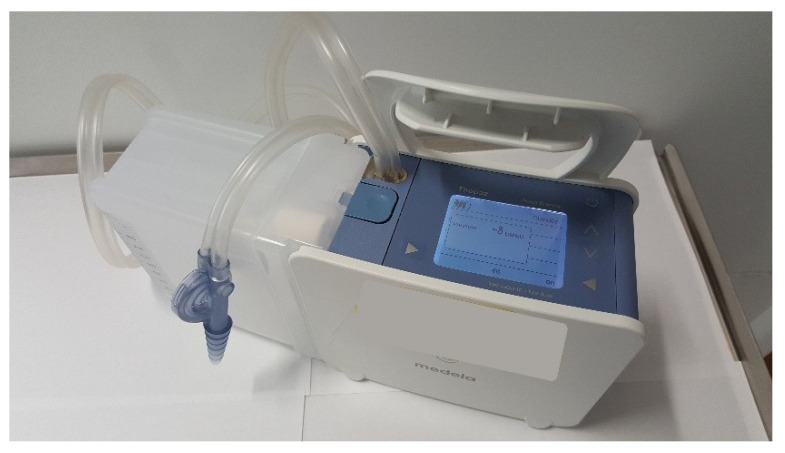
The Thopaz digital air leak drainage system.

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
