# Peer review of "Management of the Secondary Spontaneous Pneumothorax: Current Guidance, Controversies, and Recent Advances"

_jcm, 2022, doi:10.3390/jcm11051173_

Round 1

Reviewer 1 Report

Dear authors,

I found your article interesting and comprehensive. There are some small  remarks. I lectured with interest your article “Management of the secondary spontaneous pneumothorax: current guidance, controversies, and recent advances”. The article is well written and comprehensive.

There are still some remarks:

  1. In the abstract it is a little strange to make reference to a figure – figure 1, line 21, it would better fit in the introduction. It the same line, usually references are not cited in the abstract, but in the content of the article all the more so as you return to these references in the introduction
  2. In the current health context with the COVID-19 pandemic ongoing, in the part with the conservative treatment, line 88-89 you could refer more to the secondary pneumothorax associated to the infection with the SARS-CoV-2 virus, all the more so as several articles on this topic have been published,  “Martinelli, A.W.; Ingle, T.; Newman, J.; Nadeem, I.; Jackson, K.; Lane, N.D.; Melhorn, J.; Davies, H.E.; Rostron, A.J.; Adeni, A.; COVID-19 and Pneumothorax: A Multicentre Retrospective Case Series. Eur. Respir. J. 2020, 56” and references to them could complement your work.

Best regards

Author Response

Dear Reviewer

Thank you for your kind comments, which we  feel add to the article. 

  1. The Figure and reference are on Line 21, which  is situated in the introduction.
  2. Thank you highlighting this paper. The Martinelli paper is referenced in our review. We feel it is not clear from this paper about the role of conservative management in COVID patients,  as only a small proportion  were managed with a conservative pathway, and no additional information was given the patient characteristics or outcomes to support a recommendation. 

Reviewer 2 Report

Thank you for giving me the opportunity to review this manuscript and I congratulate the authors for their work.

The authors present a review of secondary spontaneous pneumothorax. They describe an overview of management of secondary spontaneous pneumothorax, such as drainage, chemical pleurodesis, autologous blood patch, endobronchial valves, and surgery. I think this review is very interesting. However, I have a minor concern, as follows:

In the section of chemical pleurodesis, it would be better to add other commonly used agents, such as glucose and OK-432.

Author Response

Dear reviewer

Thank you for your kind comments

We have added in text regarding alternate pleurodesis agents.